# Ankle and Foot Spasticity Patterns in Chronic Stroke Survivors with Abnormal Gait

**DOI:** 10.3390/toxins12100646

**Published:** 2020-10-07

**Authors:** Sheng Li

**Affiliations:** 1Department of Physical Medicine and Rehabilitation, McGovern Medical School, University of Texas Health Science Center, Houston, TX 77030, USA; 2TIRR Memorial Hermann Research Center, TIRR Memorial Hermann Hospital, Houston, TX 77030, USA

**Keywords:** spasticity, gait, stroke, botulinum toxin, nerve block

## Abstract

Chronic stroke survivors with spastic hemiplegia have various clinical presentations of ankle and foot muscle spasticity patterns. They are mechanical consequences of interactions between spasticity and weakness of surrounding muscles during walking. Four common ankle and foot spasticity patterns are described and discussed through sample cases. The patterns discussed are equinus, varus, equinovarus, and striatal toe deformities. Spasticity of the primary muscle(s) for each deformity is identified. However, it is emphasized that clinical presentation depends on the severity of spasticity and weakness of these muscles and their interactions. Careful and thorough clinical assessment of the ankle and foot deformities is needed to determine the primary cause of each deformity. An understanding of common ankle and foot spasticity patterns can help guide clinical assessment and selection of target spastic muscles for botulinum toxin injection or nerve block.

## 1. Introduction

About 80% of chronic stroke survivors have varying degrees of abnormal gait and impaired locomotion capability [1]. Spasticity in the ankle and foot muscles is very common, and often results in various ankle and foot deformities, including equinus, varus, equinovarus, and striatal toe deformities. The spastic equinovarus deformity is the most common deformity seen [2,3,4]. An ankle and foot joint abnormality could have subsequent effects on the knee, hip, and trunk position and control in post-stroke hemiplegic gait through a kinetic chain effect. For example, an equinovarus deformity shifts the ground reaction force anterior to the knee joint, and thus facilitates knee (hyper)extension during the stance phase. Stroke survivors are often forced to increase hip extension to compensate for knee (hyper)extension to keep the center of gravity within the forefoot. During the swing phase, increased knee and hip flexion is required to clear the equinovarus foot from the floor. However, they are often unable to do so due to weak hip and knee flexors, and, instead, present with hip hiking and leg circumduction. Additionally, stroke survivors have a smaller base of support due to the equinovarus deformity. The stance phase is shortened to minimize the risk of fall. Therefore, the ankle and foot deformity is often associated with kinetic and kinematic gait abnormalities, such as gait asymmetry, slow speed, genu recurvatum, etc [3,5].

Joint abnormalities in the hip, knee, ankle, and foot joints observed in post-stroke hemiplegic gait are mechanical consequences of the interactions among muscle spasticity, weakness, and disordered motor control during locomotion [6]. Depending on the severity of spasticity and weakness of muscles surrounding a joint, various joint abnormalities can develop. The complex ankle and foot anatomy contribute directly to observed deformities. As shown in Figure 1, four groups of muscles (invertors, evertors, dorsiflexors, and plantarflexors) act on the ankle–foot complex. Any isolated ankle movement is a net result of the combined activation of a group of target muscles, e.g., inversion occurs when dorsiflexors (primarily the tibialis anterior muscle) and plantarflexors (primarily the tibialis posterior muscle) co-activate. In the presence of spasticity, stroke survivors have less control and isolated activation; activation is more diffuse and divergent [7,8]. Therefore, a variety of ankle–foot deformities could be observed, depending on the severity of spasticity and weakness of individual muscles.

Among the spectrum of treatment options for ankle and foot deformities and gait disorders, interventions such as botulinum toxin (BoNT) injection and phenol neurolysis are commonly used to manage spasticity of the ankle and foot muscles [2,4,11,12]. To achieve the best clinical outcomes, it is important to identify the primary causes of the deformity. Based on the assessment from instrumented gait analysis, BoNT treatment for spasticity of target muscles has shown to improve gait pattern and walking speed [13,14,15]. However, an instrumented gait analysis lab is not available in most clinics and it is not practical to perform a computer assisted gait analysis for every patient. Understanding common ankle and foot spasticity patterns is helpful to guide our clinical assessment and development of a treatment plan. These common ankle and foot spasticity patterns are presented here through sample cases. In all cases, no significant component of contracture was detected. The common ankle and foot spasticity patterns include: equinus, varus, equinovarus, and striatal toe.

## 2. Common Deformities

### 2.1. Equinus (Plantar Flexion)

An equinus ankle and foot deformity in stroke survivors with spasticity is a condition where the ankle and foot is held in the plantarflexed position, and there is difficultly with voluntary dorsiflexion of the ankle. This is primarily caused by spasticity of the ankle plantarflexors (gastrocnemius and soleus muscles) with or without toe flexor spasticity. Figure 2 shows a 65-year-old female stroke survivor with a spastic equinus ankle and foot deformity on the right side, due to a left basal ganglia hemorrhage from hypertension three years prior. She presented with right spastic hemiplegia and abnormal gait. Her main complaint was right foot drop with difficult foot clearance during walking. This equinus deformity is evident by comparing the ankle and foot positions between the left and right sides in Figure 2A,B. During the midstance phase of the unaffected left leg (Figure 2A), the left foot has full contact with the floor, while there is foot drop on the affected right side. The toes are almost touching the floor. In contrast, during the midstance phase of the affected right leg (Figure 2B), the right heel is off the floor, while the left ankle is kept in the neutral position for foot clearance. On inspection, a callus was noted on her right forefoot, mainly located over the first metatarsophalangeal area. The modified Ashworth scale (MAS) of the ankle plantarflexor spasticity was assessed with the knee joint in both the flexed and extended positions. With the knee joint flexed to 90 degrees (Figure 2C), the ankle joint was resting at 107 degrees. MAS for the ankle plantarflexors was a 1+. With the knee joint in the extended position (Figure 2D), the ankle joint was resting at 130 degrees. MAS of plantarflexors was a 3. In both configurations, no clonus was detected, and full passive range of motion was achieved, as compared to the contralateral side. However, no active dorsiflexion was observed on the right side. No spasticity was detected for the toe flexors (flexor hallucis longus and flexo digitorum longus) or invertors (tibialis anterior or posterior muscles).

The difference in the resting ankle joint angle and MAS scores for the ankle plantarflexors with the knee joint flexed and extended revealed that the spasticity of the gastrocnemius muscle was the primary cause of the equinus deformity. The gastrocnemius muscle spans across both the knee and ankle joint, while the soleus muscle crosses the ankle joint only. Therefore, knee joint position does not affect soleus muscle spasticity. However, the contribution of the spasticity of the soleus muscle to the equinus deformity cannot be definitively ruled out. Intramuscular needle electromyography (EMG) was utilized to further evaluate muscle spasticity. Spontaneous motor unit firing has been commonly observed in spastic muscles [16,17,18]. This stroke survivor’s muscles were tested while lying on an exam bed with the right leg relaxed comfortably. Interestingly, mild spontaneous firing was present in the tibialis posterior, flexor hallucis longus, flexor digitorum longus, and tibialis anterior muscles, but not in the gastrocnemius and soleus muscles. This finding supports diffuse and divergent involvement in post-stroke spasticity [7,8]. However, when taking all of the information (history, observational gait analysis, physical exam, and EMG assessment) into consideration, the decision was made to inject botulinum toxin to the ankle plantarflexors only (150 units of incobotulinum toxin to the gastrocnemius muscle and 50 units to the soleus muscle) to correct the equinus deformity. At the six week follow-up visit, the patient reported that she walked much smoother at a faster pace than before the injection. It was observed that her right foot had full contact to the floor during the midstance phase, and she was able to dorsiflex her right ankle for foot clearance during the midswing phase. 

### 2.2. Varus (Inversion)

A varus ankle and foot deformity is a condition where the ankle and foot complex is held in the varus position, i.e., inversion. This abnormal joint position is primarily caused by coexistence of spasticity in both the tibialis anterior muscle and the tibialis posterior muscle in spastic hemiplegia. As shown in Figure 1, the tibialis anterior muscle causes the ankle dorsiflexion and inversion, while the tibialis posterior muscle causes ankle plantarflexion and inversion. The net effect of concomitant activation of these two muscles results in ankle inversion, with the ankle joint in a near neutral position, i.e., varus deformity. The varus deformity is shown in Figure 3. This patient was a 65-year-old female who sustained a left intracranial hemorrhage two years ago at an outside hospital. She presented with right spastic hemiplegia, abnormal gait, and a varus deformity. She ambulated with a point cane in her left hand. Her right ankle was fixed in the varus position during both stance and swing phases. As such, she bore her weight on the anterior lateral foot during the stance phase (Figure 3, pre-injection). A callus directly reflected the focal pressure of weight bearing on that area of the foot (Figure 4A). On physical exam, full passive range of motion into ankle dorsiflexion and eversion was unable to be achieved. Examination revealed an MAS of 3 for the tibialis anterior and tibialis posterior muscles, 2 for the plantarflexors, 1 for the flexor digitorum longus, and 0 for the flexor hallucis longus. No ankle clonus was elicited. No active ankle movement was observed in all directions. Taking history and physical exam findings into consideration, and to correct the varus deformity for better ankle and foot positioning for weight bearing, the decision was made to perform phenol (6%, aqueous solution) neurolysis. Under ultrasound and electrical stimulation guidance, motor branches of the tibial nerve to the tibialis posterior muscle (1.0 mL), medial (2.0 mL) and lateral (2.0 mL) gastrocnemius muscles, and motor branches of the peroneal nerve to the tibialis anterior muscle (2.0 mL) were blocked. As seen in Figure 3, her ankle and foot position immediately improved after phenol blocks. 

### 2.3. Equinovarus

An equinovarus deformity is the most common deformity of the ankle and foot complex in spastic hemiplegia. It is part of the “typical” hemiplegic gait pattern in stroke survivors [3,4]. This deformity is characterized by ankle plantarflexion and inversion. It is primarily caused by spasticity of the ankle plantarflexors and invertors, with minimal to no contributions from the dorsiflexors. In contrast to the varus deformity, spasticity of the tibialis anterior muscle plays a minimal role, if at all present, in the equinovarus deformity. Thus, the tibialis anterior muscle should not be the target for interventions (botulinum toxin or phenol). However, ankle plantarflexor spasticity may be present in the equinus deformity. If present, it should be addressed as demonstrated in Figure 2. 

Figure 5 shows a stroke survivor with an equinovarus deformity before and after phenol injection. This 34-year-old patient had an intracranial hemorrhagic stroke four years ago. He presented with right spastic hemiplegia, abnormal gait, and an equinovarus deformity. He ambulated with a quad cane. During the stance phase of the affected right leg, his right heel was off the floor and the ankle was inverted (Figure 5A). He bore his weight on the right lateral forefoot, as manifested by a callus in that area (Figure 5C). He had full passive range of motion of his right ankle. MAS was a 3 for the ankle plantarflexors, 2 for the invertors (the tibialis posterior muscle), 0 for the flexor hallucis longus and flexor digitorum longus, and 0 for the tibialis anterior muscles. No clonus was elicited. He had weak voluntary ankle plantarflexion, but no active dorsiflexion was detected. Under ultrasound and electrical stimulation guidance, motor branches to the medial (1.5 mL) and lateral (1.5 mL) gastrocnemius, soleus (1.0 mL), and tibialis posterior (0.5 mL) muscles were blocked with 6% phenol (aqueous solution). His ankle and foot position immediately improved after the injection (Figure 5B, post-injection). 

### 2.4. Striatal Toe

The striatal toe deformity describes a condition of hyperextension of the great toe (Figure 6). It is seen in patients with dystonia, Parkinson’s disease, and spastic hemiplegia after stroke [19,20]. The striatal toe deformity is primarily caused by spasticity of the extensor hallucis longus (EHL) in stroke survivors. Botulinum toxin (BoNT) treatment has been effective, but the dose and accompanied involvement of other muscles are variable [19,20]. Though it is easy to recognize this deformity and to treat it with BoNT, the clinical outcome depends on the dose and treatment of accompanied spastic muscles. The case in Figure 6 is a good example. 

This was a 64-year-old female with a medical history of a patent foramen ovale and right middle cerebral artery stroke 15 months prior, with residual left hemiplegia and abnormal gait. She presented with a one-year complaint of hyperextension of her left great toe with associated pain. She regained the ability to walk several weeks following her stroke with subsequent development of great toe hyperextension. She had two previous injections to her left extensor hallucis longus (EHL) (50 and 100 units of Onabotulinum toxin A, respectively) at an outside facility without improvement. The patient was referred to our clinic for possible phenol neurolysis for EHL spasticity. 

On exam, the patient had hyperextension of her left great toe of ~60 degrees in a resting supine position (Figure 6, pre-injection). Walking with a cane, she initially had full foot contact on the left affected side during the mid-stance phase, but the left ankle then became plantarflexed (~30 degrees) and the great toe hyperextended after a few steps. Needle electromyography showed spontaneous firing greatest in the left gastrocnemius and EHL muscles, with moderate activity of the soleus muscle and faint activity of the tibialis anterior muscle. A diagnostic lidocaine block of her tibial nerve motor branches to the gastrocnemius and soleus muscles was performed, resulting in full foot contact with the floor and greatly reduced great toe hyperextension during the stance phase. The patient was then scheduled for incobotulinum toxin injections to the left gastrocnemius (75 units), soleus (50 units), and EHL (25 units). Two months after injections, she exhibited full foot contact with the floor without great toe extension during the mid-stance phase, with concomitant reduction of pain and improved walking speed (Figure 7).

## 3. Discussion

The above ankle and foot deformities are commonly seen in clinical practice. These common spasticity patterns usually present with different levels of severity. Proximal knee and hip muscles also have different degrees of weakness and spasticity that may alter the ankle and foot deformities over time. In addition, other rare presentations may be seen, for example, focal spasticity of the extensor digitorum longus or toe curling from intrinsic foot muscle spasticity [21]. Therefore, various clinical presentations of ankle and foot spasticity are observed in the presence of a spastic hemiplegic gait. Despite a general presentation of extensor spasticity in the lower extremity, there is no typical ankle and foot deformity in post-stroke hemiplegia. Similar to various spastic hemiplegic gait patterns, this versatility of ankle and foot spasticity involvement reflects that ankle and foot deformity is a mechanical consequence between muscle spasticity, weakness, and its interactions with the ground reaction forces [6]. 

The standard algorithm to treat ankle and foot deformities and gait disorders can be found elsewhere [2,11]. Understanding common ankle and foot spasticity patterns is helpful to guide clinical assessment and selection of target spastic muscles for botulinum toxin injection or nerve block. However, contracture should be identified first, since, if present, different treatment plans are necessary [2,11]. Comprehensive and careful assessment is needed to determine the primary cause of each deformity. In addition to routine history and physical exam including observational gait analysis, EMG and diagnostic lidocaine blocks may be required to make an accurate diagnosis. It is ideal to have instrumented gait analysis, but this is not practical for most clinics. EMG findings of spontaneous motor unit firing should be used with caution. These findings suggest the presence of spasticity in a muscle. However, EMG findings do not explain how or whether spastic muscles contribute to abnormal ankle and foot position or gait patterns. In the equinus deformity seen in Figure 2, spontaneous motor unit firing was diffusely present, which is not uncommon [8]. However, history and physical exam suggested the gastrocnemius and soleus muscle spasticity was the primary contributor to the gait disorder. Lidocaine blocks should be diagnostic and confirmatory of the clinical observations (Figure 7).

Understanding the different ankle and foot spasticity patterns and their underlying primary causes is helpful to guide the intervention plan. Tibialis anterior muscle spasticity is a primary contributor of the varus deformity, while its role is negligible in the equinovarus deformity. However, management of tibialis anterior muscle spasticity has been controversial. Targeting this muscle is included in the treatment decision-making algorithm by one group of clinicians [2], but it is not considered by other groups [11,22]. As demonstrated in Figure 3, successful suppression of tibialis anterior muscle spasticity contributes to correction of the varus deformity. However, successful correction of the equinovarus deformity does not need to address spasticity of the tibialis anterior muscle. These two contrasting examples reveal that whether the tibialis anterior muscle spasticity needs to be managed depends on the severity of spasticity and its mechanical consequences. If spasticity is severe, sustained involuntary activation of this muscle causes ankle dorsiflexion and inversion. The net result is a varus deformity in the presence of tibialis posterior muscle spasticity. Apparently, tibialis anterior muscle spasticity is not severe enough to counterbalance the plantarflexion activity from the tibialis posterior muscle spasticity in the equinovarus deformity. If uncertain, lidocaine diagnostic block is helpful.

It is important to understand the mechanical consequences of spastic muscles during functional tasks, i.e., gait. The striatal toe deformity (Figure 7) is another prime example. Great toe hyperextension is caused by spasticity of the extensor hallucis longus (EHL). Only treating the EHL spasticity was not adequate to solve the problem. The fact that the patient’s heel was elevated off the floor after a few steps suggests that EHL spasticity may develop gradually to improve ankle and foot stability during the stance phase in order to compensate for plantarflexor spasticity and its mechanical consequence of an equinus deformity. It is evident by the fact that a lower dose of BoNT was sufficient to suppress the EHL spasticity, with concomitant treatment of plantarflexor spasticity (Figure 7). It has been reported that co-treatment of the plantarflexors and EHL are needed in some striatal toe cases [19,20].

## 4. Conclusions

Clinical presentations of ankle and foot spasticity are various. There are mechanical consequences of interactions between spasticity and weakness of surrounding muscles during walking. Careful and thorough clinical assessment of the ankle and foot deformities is needed to determine the primary causes of each deformity. Understanding common ankle and foot spasticity patterns is helpful to guide clinical assessment and selection of target spastic muscles for botulinum toxin injection or nerve block.

## Figures and Tables

**Figure 1 toxins-12-00646-f001:**
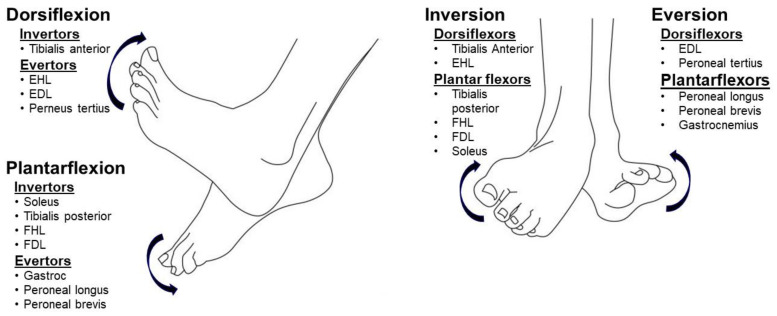
The muscles involved in ankle movement. EHL: extensor hallucis longus; EDL: extensor digitorum longus; FHL: flexor hallucis longus; FDL: flexor digitorum longus; Gastroc: gastrocnemius. Note: Tendons of the soleus and gastrocnemius merge and form the Achilles tendon. The soleus tendon is located in the medial portion of the Achilles tendon, thus contributing more to inversion, while the gastrocnemius tendon forms the lateral portion and contributes more to eversion. They plantarflex the ankle joint when acting together [9,10].

**Figure 2 toxins-12-00646-f002:**
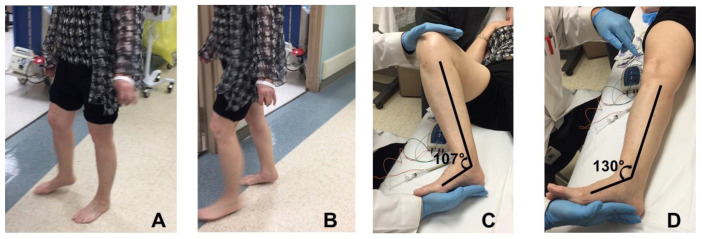
(**A**) Stroke survivor with an equinus ankle and foot deformity on the right side. (**A**): Left leg in the midstance phase, (**B**): right leg in the midstance phase, (**C**): the resting ankle joint with the knee joint flexed, and (**D**): the resting ankle joint with the knee joint fully extended.

**Figure 3 toxins-12-00646-f003:**
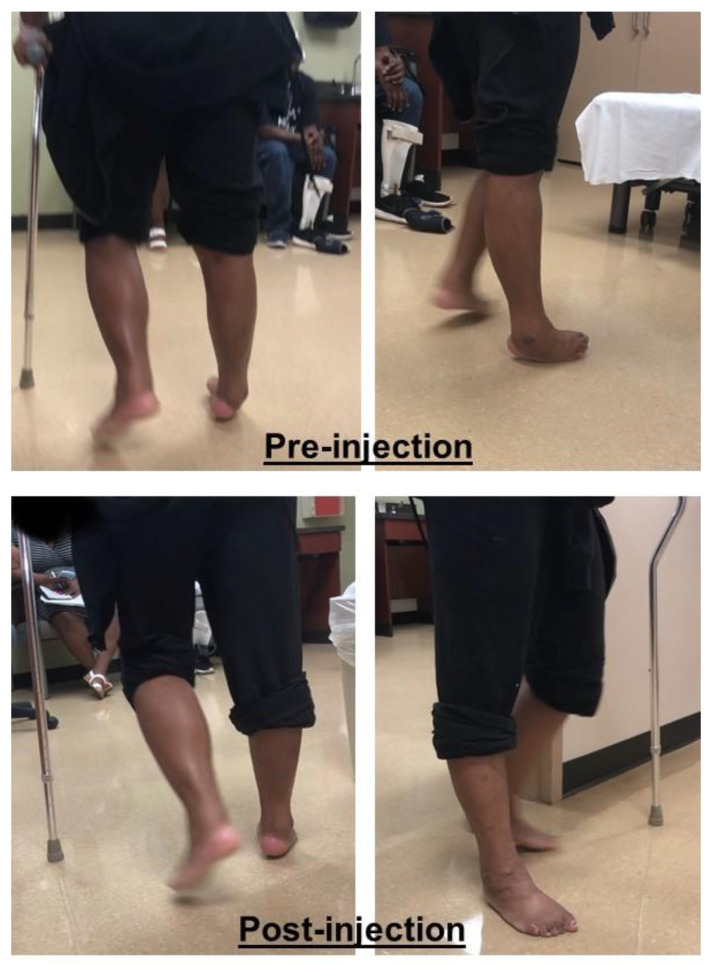
A stroke survivor with a varus deformity. Her foot and ankle position improved immediately after phenol blocks.

**Figure 4 toxins-12-00646-f004:**
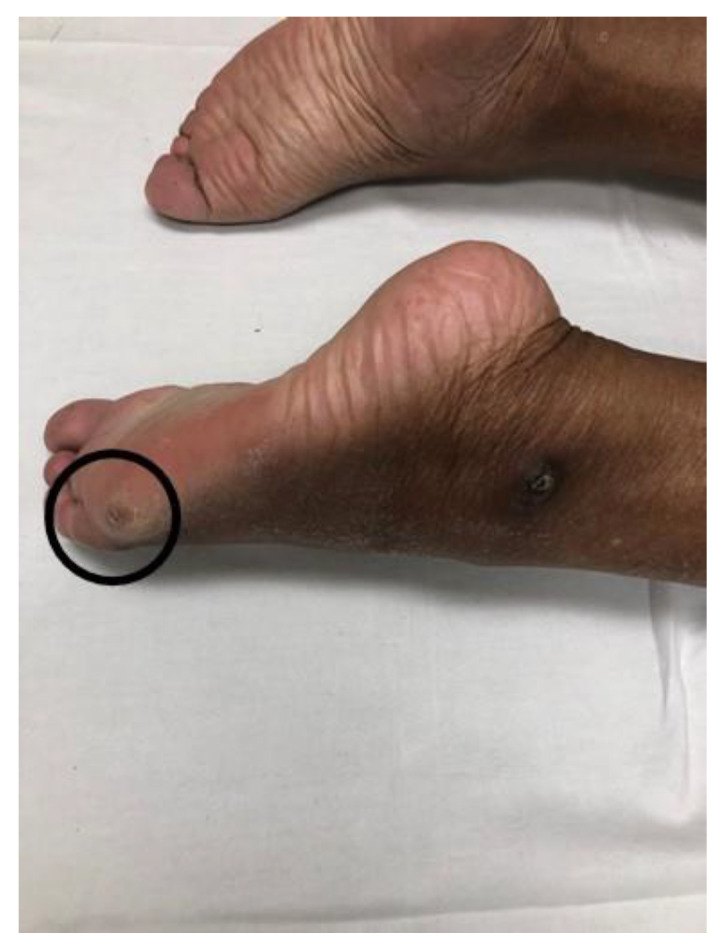
The callus formation on the anterior lateral board of the foot with a varus deformity.

**Figure 5 toxins-12-00646-f005:**
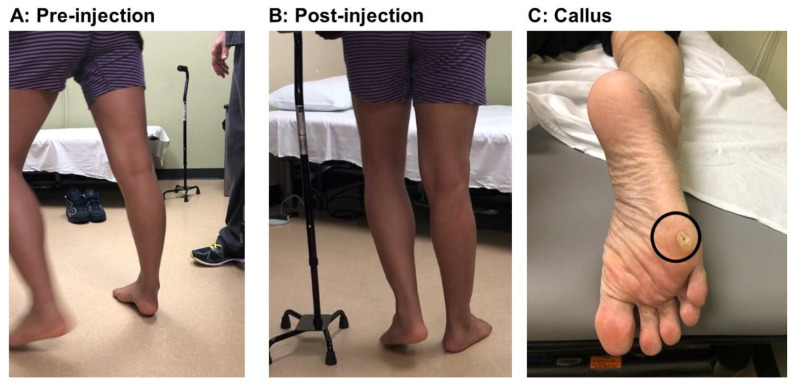
A stroke patient with right spastic hemiplegia and equinovarus deformity during the mid-stance phase before (**A**) and after (**B**) phenol injection. Note, the callus formation is in the lateral forefoot (**C**) in the equinovarus deformity.

**Figure 6 toxins-12-00646-f006:**
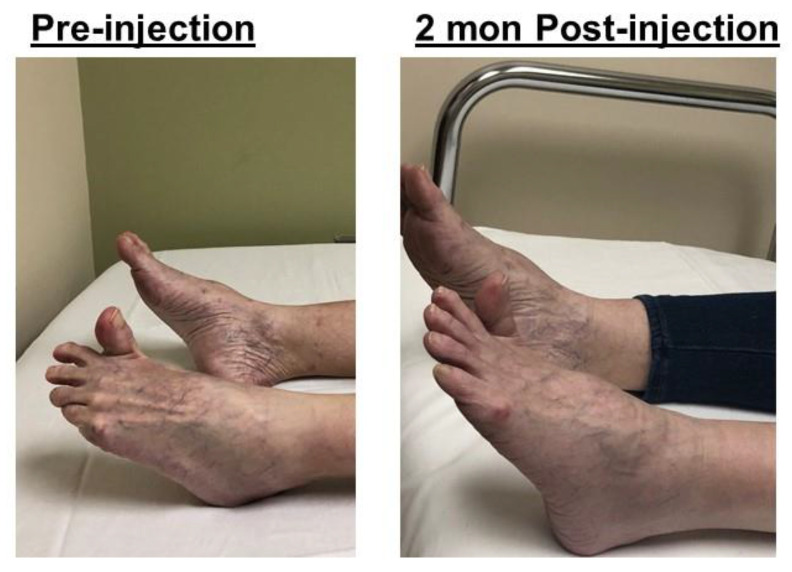
The striatal toe before and two months after botulinum toxin injection.

**Figure 7 toxins-12-00646-f007:**
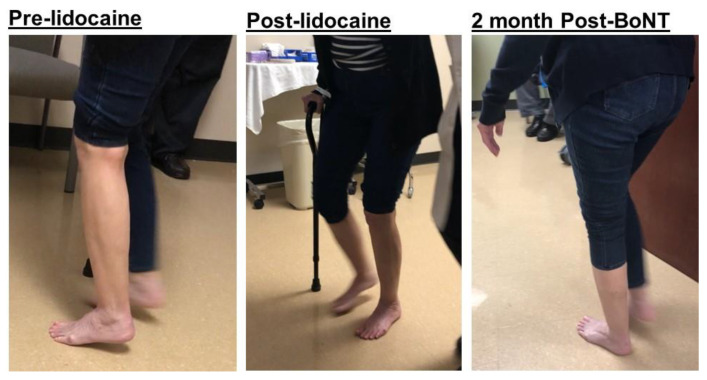
A stroke survivor with a left striatal toe has ankle plantarflexion during the mid-stance phase to compensate for great toe hyperextension (**left**). After a successful lidocaine block of the tibial nerve motor branches (**middle**), she received botulinum toxin (BoNT) to the gastrocnemius, soleus, and extensor hallucis longus (EHL) with a successful outcome (**right**).

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
