# Peer review of "Ankle and Foot Spasticity Patterns in Chronic Stroke Survivors with Abnormal Gait"

_toxins, 2020, doi:10.3390/toxins12100646_

Round 1

Reviewer 1 Report

Great range of ankle and food muscle spasticity presentation in chronic stroke was reviewed. It is an important review and summary for clinical translation. Tough some aspect of the primary cause is discussed, including some pathophysiology will help to understand the manifestations and the efficacy of BTOX. Some changes seen after BOTOX injection should be characterized and presented in more detail.

Author Response

RE: Thank the Reviewer for the encouraging comments. Detailed information of medication (BoNT and phenol) is now provided for all cases.  It is unfortunate that a post-injection photo for the first case was not available. According to the medical records, I added more details in the revised manuscript for the first case.

“the decision was made to inject botulinum toxin to the ankle plantarflexors only (150 units of inco-botulinum toxin to the gastrocnemius muscle and 50 units to the soleus muscle) to correct the equinus deformity.  At the 6 week follow-up visit, patient reported she walked much smoother at a faster pace that before injection. It was observed that her right foot had a full contact to the floor during the midstance phase, and she was able to dorsiflex her right ankle for foot clearance during the mid-swing phase.”

Reviewer 2 Report

This is an interesting manuscript dealing with 4 patients who suffered from stroke and who have clinical consequences in their feet and ankles. Also the authors report effective treatment for these sequelae. However, I have major concerns that, in my opinion, should be taken into account before acceptation for publication.

Lines 21 to 34, the authors state that “Spasticity in ankle and foot muscles is very common, and often results in various ankle and foot deformities”. However, they spend too many lines detailing the gait patterns of only one of the deformities (equinovarus). The authors should include explanation of all the deformities mentioned.

Lines 49-51, the authors sustain that soleus muscle is invertor, and gastrocnemius is evertor, and they add a reference (no. 9) dated in 1988. More recent studies have demonstrated that gastrocnemius may play a role in both movements, inversion and eversion, as well as soleus. These concepts must be reviewed and modified.

Line 65, please correct “dosriflex”.

Line 102, the authors state that the patient treated showed a great response in the 6 week followup visit thanks to the treatment applied. It would be helpful if some data or images of the 6 week followup visit are included in order to show changes (as done in figure 3).

Line 149, please correct “positionimmediately”

Author Response

This is an interesting manuscript dealing with 4 patients who suffered from stroke and who have clinical consequences in their feet and ankles. Also the authors report effective treatment for these sequelae. However, I have major concerns that, in my opinion, should be taken into account before acceptation for publication.

 Lines 21 to 34, the authors state that “Spasticity in ankle and foot muscles is very common, and often results in various ankle and foot deformities”. However, they spend too many lines detailing the gait patterns of only one of the deformities (equinovarus). The authors should include explanation of all the deformities mentioned.

 RE: The gait patterns of the equinovarus deformity was described here as an example. The purpose was to illustrate the subsequent effects of the ankle and foot deformity through a kinetic chain effect, thus highlight the importance of the ankle and foot deformity.

It would be too lengthy in the Introduction to describe gait patterns after all ankle and foot deformity. To highlight this is only an example, all deformities are listed early in the statement, see below.

“Spasticity in the ankle and foot muscles is very common, and often results in various ankle and foot deformities, including equinus varus, equinovarus, and striatal toe deformities.”

Lines 49-51, the authors sustain that soleus muscle is invertor, and gastrocnemius is evertor, and they add a reference (no. 9) dated in 1988. More recent studies have demonstrated that gastrocnemius may play a role in both movements, inversion and eversion, as well as soleus. These concepts must be reviewed and modified.

RE: Thanks for insightful comments. I have updated the references with more recent studies. Also, the statements have been revised to be more accurate, as the Reviewer suggested. See below:

“Tendons of the soleus and gastrocnemius merge and form the Achilles tendon. The soleus tendon is located in the medial portion of the Achilles tendon, thus contributing more to inversion, while the gastrocnemius tendon forms the lateral portion and contributes more to eversion. They plantarflex the ankle joint when acting together.”

  1. Arndt, A.; Brüggemann, G.P.; Koebke, J.; Segesser, B. Asymmetrical loading of the human triceps surae: I. Mediolateral force differences in the Achilles tendon. Foot and Ankle International 1999,20,444-9.
  2. Bojsen-Møller, J.; Magnusson, S.P. Heterogeneous Loading of the Human Achilles Tendon In Vivo. Exercise and Sport Sciences Reviews 2015,43,190-7.

Line 65, please correct “dosriflex”.

RE: Thanks for careful reading. The typo has been corrected.

Line 102, the authors state that the patient treated showed a great response in the 6 week followup visit thanks to the treatment applied. It would be helpful if some data or images of the 6 week followup visit are included in order to show changes (as done in figure 3).

RE: It is unfortunate that a post-injection photo for the first case was not available. According to the medical records, I added more details in the revised manuscript for the 6-week follow up visit for the first case.

 “the decision was made to inject botulinum toxin to the ankle plantarflexors only (150 units of inco-botulinum toxin to the gastrocnemius muscle and 50 units to the soleus muscle) to correct the equinus deformity.  At the 6 week follow-up visit, patient reported she walked much smoother at a faster pace that before injection. It was observed that her right foot had a full contact to the floor during the midstance phase, and she was able to dorsiflex her right ankle for foot clearance during the mid-swing phase.”

Line 149, please correct “positionimmediately”

RE: Thanks for careful reading. It has been corrected.

Reviewer 3 Report

The authors presented a cursory display of various clinical patterns in foot and ankle deformities following stroke via 4 different clinical cases. In any case, there are no clear statements about the exclusion of fixed or partially fixed deformities which may be the case in long-standing paresis or spasticity and which might interfere with any injection therapy.

In the present form, the statements are rather propedeutic and are of little clinical value as the treatment options with botulinum toxins or phenols do not become clear for the reader. Therefore, the authors should add their standard algorithm how they treat these passively correctable foot and ankle deformities. Further, any information about the repetitive treatment is missing, as well as the final outcome.

Do you have any information about your results in this specific pathology in larger population apart from these 4 cases?

Summarizing, the manuscript in its present form has to many shortcomings. Publication cannot be recommended.

Author Response

RE: The primary goal is to emphasize that various ankle and foot deformities in hemiplegic stroke survivors are mechanical consequences of interactions between spasticity and weakness of surrounding muscles during walking. Different deformities are discussed through sample cases. For example, different approaches in managing tibialis muscle spasticity between two treatment algorithms[2,12] are discussed. Understanding these common presentations can help guide clinical assessment and selection of target spastic muscles for botulinum toxin injection or nerve block.

It has been made clear regarding the standard treatment algorithm and the importance of understanding common spasticity patterns, see below.

 “The standard algorithm to treat ankle and foot deformities and gait disorders can be found elsewhere[2, 12]. Understanding common ankle and foot spasticity patterns is helpful to guide clinical assessment and selection of target spastic muscles for botulinum toxin injection or nerve block. However, contracture should be identified first, since, if present, different treatment plans are necessary [2, 12]. “

  1. Deltombe, T.; Wautier, D.; De Cloedt, P.; Fostier, M.; Gustin, T. Assessment and treatment of spastic equinovarus foot after stroke: Guidance from the mont-godinne interdisciplinary group. Journal of Rehabilitation Medicine 2017,49,461-8.
  2. Nonnekes, J.; Benda, N.; Van Duijnhoven, H.; Lem, F.; Keijsers, N.; Louwerens, J.W.K. et al. Management of gait impairments in chronic unilateral upper motor neuron lesions a review. JAMA neurology 2018,75,751-8.

My understanding is that results from larger sample size are not necessary for this type of clinical communication or perspective articles.  

Round 2

Reviewer 2 Report

I think adequate responses have been given to the comments. Thanks.

Reviewer 3 Report

The manuscript reports about 4 cases with a moderate in-depth analysis of the neurologic situation regarding muscular strength and spasticity of the foot and ankle following stroke. The clinical value for the readership is limited as the authors did not reveal their general approach and algorithm of treatment. The authors' reply after the first review did not add any gain in content.